# Juggling during Lockdown: Balancing Telework and Family Life in Pandemic Times and Its Perceived Consequences for the Health and Wellbeing of Working Women

**DOI:** 10.3390/ijerph20064781

**Published:** 2023-03-08

**Authors:** Mariana Loezar-Hernández, Erica Briones-Vozmediano, Elena Ronda-Pérez, Laura Otero-García

**Affiliations:** 1Department of Nursing and Physiotherapy, Faculty of Nursing and Physiotherapy, University of Lleida, 25008 Lleida, Spain; 2Consolidated Research Group Society, Health, Education and Culture (GESEC), University of Lleida, 25008 Lleida, Spain; 3Research Group of Health Care (GRECS), Biomedical Research Institute (IRB), 25003 Lleida, Spain; 4Department of Community Nursing, Preventive Medicine, Public Health and History of Science, University of Alicante, 03690 Alicante, Spain; 5Consortium for Biomedical Research in Epidemiology and Public Health (CIBERESP-ISCIII), 28029 Madrid, Spain; 6Department of Nursing, Faculty of Medicine, Universidad Autónoma de Madrid, 28049 Madrid, Spain

**Keywords:** work-life balance, work-family conflict, COVID-19, telework, women’s health, gender role, qualitative research

## Abstract

The COVID-19 pandemic disrupted work-family balance due to lockdown measures. The aim of this study was to explore the experiences of working mothers in Spain and the consequences of trying to balance work and family for their health and wellbeing. We conducted a qualitative study based on 18 semi-structured interviews with mothers of children under 10. Five themes were identified: (1) Telework—characteristics and challenges of a new labor scenario; (2) Survival and chaos—inability to work, look after children, and manage a household at the same time; (3) Is co-responsibility a matter of luck?—challenges when sharing housework during lockdown; (4) Breakdown of the care and social support system; and (5) decline in health of women trying to balance work and family life. Mothers who had to balance telework against family life suffered physical, mental, and social effects, such as anxiety, stress, sleep deprivation, and relationship problems. This study suggests that, in situations of crisis, gender inequality increases in the household, and women tend to shift back to traditional gendered roles. Governments and employers should be made aware of this, and public policies should be implemented to facilitate work-family reconciliation and co-responsibility within couples

## 1. Introduction

As a result of the COVID-19 pandemic, the Spanish government imposed stay-at-home mandates to reduce people’s movements, the risk of infection, and the spread of the disease [1,2]. These measures included the closure of all educational institutions and a transition from in-person work to telework: the change was implemented by 48.8% of businesses, as opposed to 14.8% before the pandemic [3]. Working from home entailed changes in productivity standards, more intense work, and longer hours, the expectation that employees should be available outside work hours, and difficulties separating personal life from work [4,5,6,7,8]. This exacerbated work-family conflicts, especially in the case of women, because of the increase in domestic responsibilities, while men were subject to greater work demands [9]. Women pursued different strategies to confront this problem, including, for example: radically reorganizing home spaces to be able to work; constantly negotiating with their children for the use of technologies; modifying routines and work hours; and using paid leave days to cover their children’s needs, given the difficulties in parenting and working at the same time [10,11]

The pandemic aggravated inequalities in the division of housework and childcare, due to the absence of resources such as family support or external help [8,12]. The need for these support structures became evident during the lockdown, when people were forbidden from interacting with others, except the nuclear family [13]. In Spain, 70% of domestic work and childcare was undertaken by women; co-responsibility within couples remained unchanged in 66% of households and worsened in 13% of households [14,15]. According to Hochschild [16], it is mostly women who take on the “second shift” of unpaid childcare and housework, and the lack of work-family reconciliation policies remain an impediment to achieving gender equality. Women’s domestic workload is not only a physical effort, but also a mental one, due to the responsibility of planning tasks and organizing the household [8,12]. In turn, women’s satisfaction with how they share tasks with their partners acts as a protective factor against stress and mental health effects [17].

Previous international studies [7,8,18] have shown that the work-family conflict impacts women’s health, with effects such as anxiety, distress, fatigue, exhaustion, lack of energy, stress about their work situation and even feelings of guilt. In addition, there were consequences regarding lifestyle, including inadequate physical exercise and an increase in alcohol consumption [18]. Indeed, in a German study [11], mothers of small children were identified as a particularly vulnerable group during the COVID-19 pandemic.

Studies conducted in Spain [19,20] have confirmed that the pandemic had greater psychological consequences for women compared to men. This is because women became more stressed, had greater conflicts from balancing work and family, were more afraid of the disease and worried more about their precarious work situations. These studies also show that work-related insecurity and the work-family conflict are related to higher levels of anxiety, depression, insomnia and personal and work conflicts [20,21,22]. However, no studies have delved into the perception of working mothers of small children who had to balance work responsibilities and childcare during the COVID-19 pandemic. Accordingly, we conducted this study to explore the experiences of working mothers in Spain and the consequences of trying to balance work and family for their health and wellbeing.

## 2. Materials and Methods

### 2.1. Study Design

This was a phenomenological qualitative study, conducted in Spain between June 2021 and September 2022, with an interpretative/constructivist approach [23]. Phenomenology contributes to a deep understanding of lived experience, as it aims to capture its meaning from the participants’ perspectives [24,25]

### 2.2. Participants

A total of 18 women were identified by purposive sampling. The mean age of the women was 41 years (range 35–54 years), they were employees in both the private and public sectors, and they worked mostly in full-time positions (Table 1). As per the inclusion criteria, we recruited heterosexual women who were living with their husbands or partners, had children below the age of 10, and teleworked during the COVID-19 lockdown (15 March to 21 June 2020). Participants were referred to us by four key informants in a previous study [26] and subsequently, by snowball recruiting.

### 2.3. Data Collection

We conducted semi-structured interviews using a script designed according to the study objectives and a literature review (Table 2). We asked women about their experience working from home, balancing work and family, reorganizing the division of childcare and domestic tasks, and their self-perceived health. Because of movement limitations during the lockdown, the interviews were conducted by video call or telephone and lasted between 60 and 90 min. The interviews were recorded and then transcribed, along with field notes, to collect the interviewees’ emotional lability, among other aspects.

### 2.4. Data Analysis

Two of the authors separately analyzed the information contained in the transcriptions, following the six phases of reflexive thematic analysis proposed by Braun and Clarke [27,28]: (1) The first author conducted most of the interviews, so she performed the first step of reading and re-reading the transcriptions to become familiarized with the data, taking notes of pre-analytical ideas in memo format; she then (2) systematically encoded the interviews and grouped codes according to similarities; (3) the code groups were pooled together in order to undertake a common search for initial candidate themes; (4) common candidate themes were then reviewed and refined in a process of continuous feedback using the original data, leading to a common preliminary theme structure; (5) the argument within each theme were then described and defined, leading to the final naming of the themes, as agreed on by the two researchers and further discussed within the team; (6) finally, the final themes and categories were organized and chosen by the first author, using the most representative verbatim quotes for each theme. This final structure was sent to all team members to verify their agreement on the final version and interpretation of the data. The ATLAS.ti software (v. 9) was used throughout this process as a tool to support the organization and coding of the information during the analysis.

### 2.5. Rigor and Reflexivity

To enhance reflexivity and rigor, we developed a reflexive process in the design, collection and analysis of the data, and in the writing of this final report. We also formulated memos after conducting, listening to, and transcribing the interviews [29]. The two researchers who collected and analyzed the data had an outsider perspective on the situation analyzed in the study, because they had not faced this work-life balance issue during the strict phase of the lockdown. This position provided a fresh and different viewpoint in the analysis; nonetheless, because of the theoretical limitations of an outsider perspective, these two researchers discussed their pre-analytical ideas and result structures with the rest of the team, who had an insider perspective, as they were also working mothers.

### 2.6. Ethical Considerations

This study was approved by the Ethics Committee at Hospital Arnau de Vilanova (CEIC-2460), Lleida, Spain. The participants’ real names were replaced by pseudonyms, in order to preserve their confidentiality, and verbatim accounts do not include any information that could be used to identify them.

## 3. Results

Table 3 shows the main themes and categories identified, along with the codes selected for each one.

### 3.1. Telework—Characteristics and Challenges of a New Labor Scenario

The participants explained that telework was implemented by companies during the lockdown in a haphazard fashion. Many of them were facing this situation for the first time, as a result of the COVID-19 emergency. Working from home, women reported an increased demand for immediate availability and uninterrupted connection to work, even outside their normal working hours. This was linked, in some cases, to the lack of physical presence—women were expected to demonstrate to employers that they were, in fact, working.


*And, on top of all that, you were expected by your employer to be available 24/7. They could call you at any time. It didn’t bother them. At whatever time they felt like—it seemed like you had to be on call all the time.*
(Ana)


*The problem with working from home is that it’s not regulated. It’s like you have to be on line 24 h a day, 7 days a week.*
(Clara)

Women consistently reported that working from home during the lockdown was not the experience they had expected. This was mainly due to the presence of their children at home and the organizational challenges this posed with their partners. As a result, the disadvantages and negative aspects of working from home emerged more easily in the participants’ accounts. These include: difficulties setting boundaries between work and family life; a loss of routines and interactions with colleagues; and la ack of confidence with their work, as they could not resolve issues with other colleagues. In some cases, this led to feelings of loneliness and lack of motivation in their work.


*Well, one thing is working from home when there’s nobody else there, and you can focus, and another thing is having to work from home because you’re forced to stay in with the kids. That’s not working from home, that’s not the real “remote work,” because you don’t get a certain amount of time to be on line working and showing what you can and cannot do.*
(Ana)


*There were some advantages, or at least what I see as advantages, but there was also the downside of “When do I stop?” Right? When do I really switch off and leave? That’s what happens when you work in a real office—you just turn off your computer and go home. But now your home is your office in addition to your home. So, you don’t do it.*
(Emilia)

In addition, women did not have segregated areas at home that they could use exclusively for professional purposes. They had to share their workstations with their partners and children, which affected their concentration and prompted interruptions. The distribution of home spaces with the women’s partners was variable. However, it was mostly men who occupied the most secluded spaces, which allowed them to work under better conditions.


*My husband started to work at the baby’s changing table. Well, our baby’s changing table is quite a big table, so he used that as his office, and I was in the living room. On days when we both had to work in the living room, then we both sat there, on either side of the dining table. That’s all the space we have.*
(Marta)


*Well, they sent [the laptops] to us by mail. Basically yes, he did have to work from home, he had no option. He went into a quiet room so he could carry on working.*
(Emilia)

However, Mercedes considered that working from home is just another way of working—one that does not necessarily mean better work-family balance.


*Working from home doesn’t mean work-family balance; it just means working in your house. That doesn’t mean that you can balance your work and family more easily, like you can take care of the children. You’re working, you’re not with the children. You can either do one thing or the other. They sold us this idea that working from home promotes work-family balance. I don’t think this is the perfect solution—working from home is not how you balance your work and family life.*
(Mercedes)

As perceived by the women, employers showed a relentless interest in returning to in-person work, regardless of how that would impact the women’s organization of their households.


*They don’t like [telework]. For example, the minimum staff was kept here in the office. Almost everyone did work from home, but then in June people started to come back and I was also pressured into coming back. And I was like: “But I’ve got the children…”*
(Marta)

When schools reopened after the strictest part of the lockdown, women easily identified some advantages of telework in terms of balancing work and family. These included, among others, saving commute time and finding it easier to pick up children from school.


*Working from home because of the pandemic has many good things, because it helps you balance your life… You can take them to school, come back… I mean, it’s me who does all the parenting. That’s been a clear advantage.*
(Carla)


*There’s a lot of positive things about being at home. When you have two kids who take up a lot of your time, you’re more relaxed, because you can sleep longer. We save two hours by not commuting to work.*
(Cristina)

### 3.2. Survival and Chaos—Inability to Work, Look after Children, and Manage a Household at the Same Time

#### 3.2.1. Constantly Combining Work and Childcare within the Same Physical and Temporal Space

“Survival” is a term used by the women in our study to describe their daily lives during the lockdown. What began as an idyllic situation, in which they could spend more time with their families, ended up being catastrophic, as some of them reported. This was primarily because they had to prioritize between their children’s needs and their own professional duties, which caused tension and stress.


*But the first few days were wonderful, because you have a lot more time to take care of the house. It seemed idyllic. But if that drags on, it becomes catastrophic, because it went on for so long! Work ended up taking up almost my whole day, and then I had to juggle to get everything done.*
(Cristina)


*Actually I just survived. I didn’t have time to get organized. I was trying to cover basic needs, getting the girls dressed. I say “dressing” because otherwise it meant leaving them in their pajamas all day. And then you have to feed them breakfast and leave them plugged into something, right? Surviving was the only option.*
(Sara)

The women’s routine during the lockdown was a constant combination between work and family; for them, it was practically impossible to separate the two. Because of this imbalance, women recognized their failure to comply with the expectations and self-demands of their roles as mothers and professionals. This failure engendered feelings of guilt and frustration in some of the interviewees.


*You don’t get either the time or the focus you normally need for work. You’re constantly combining things. Now you come in, now I go out in an hour, but then at one o’clock I need to log onto a meeting. Now you get a phone call, you say “OK, I’ll pick up.” But I’m with the kids, so when the kids realize you’re on the phone, they go completely wild.*
(Eva)


*During the day, zero balance. I couldn’t. I just… I was not a good mom or a good employee. Nothing, not at all. I don’t know how other people see it. There was no balance for me.*
(Inés)

In fact, Ana experienced feelings of resignation when she had to establish priorities and lower her expectations.


*Work-family balance is a bit like finding an agreement, saying “OK, come on.” I mean, I’m not from here, and I am not Superwoman, and my husband doesn’t have to be a Superman. Clearly you do as much as you can. Keeping the bathroom nice and clean, well, even if the bathroom and kitchen are clean, then, if you can’t make the beds because one day there’s no time to make the beds, well, you just don’t make the beds.*
(Ana)

#### 3.2.2. Strategies Adopted to Survive the Lockdown Routine

*Working more: spartan routines to combine childcare and work.* Women and their partners set up shifts so they could work and take care of their children at the same time. This resulted in longer work days. In critical cases, some women worked 12 to 14 h days.


*“What you can’t do is get up at 5 o’clock in the morning to work” [the doctor would tell her] and I said “OK, yes, but it’s the only way we can get by.” One of us gets up at dawn while the other one stays up all night… Otherwise, nobody can work with a little girl at home.*
(Nerea)


*Well look, working from home during the lockdown was real madness, I mean, damn crazy. I’d start work before my normal start time. I’d get up at 6 a.m. or 6:30 at the latest, and I’d start to work to catch up… I was returning phone calls till really late.*
(Ana)

*Complicity among coworkers*. A sense of comradeship developed among coworkers while working from home. Complicity arose as all women were experiencing similar situations and challenges. One such challenge was coordinating work schedules among female coworkers. Women changed their working hours to facilitate childcare; however, their hours were not always synchronized with their coworkers’ schedules, which created difficulties for teamwork.


*We were all going through more or less the same. I’d have my daughter sitting beside me, alright, but this guy on the phone there would have his three kids jumping and screaming around him. In the end, we all understood people’s situations. We were all working but we knew that, at the same time, we were stirring a lentil stew, or doing whatever it was time to do.*
(Consuelo)


*I was working very early hours. Some other people were working in the evening. We were never forced to coordinate. We weren’t asked to work from such time to such time. It was up to everyone’s responsibility really. So, if they let me do as I thought best, then the least I could do was to be there for them if they needed me.*
(Nerea)

*Setting strict boundaries to work as a means of survival.* The women’s boundaries between their work and their personal lives became blurred when working from home, mainly because all their activities were performed within the same physical space. Transitions between the work sphere and the personal sphere, such as commuting, which had often served as spaces for resting and switching off, vanished.


*Some people might have coped very well, but for me it was untenable.*
(Nerea)


*The bad thing is that you don’t even get your commute back home [laughing], whether it’s short or long. As soon as I walk out this door, the children begin asking for this, crying for that […]. You walk out of your room, and you’re done with work. There is no transition.*
(Carla)

In this scenario, women needed to establish strict boundaries in their work and refused to satisfy some of their employers’ demands. Setting boundaries was seen as a strategy for survival and self-protection in an adverse scenario marked by an overload of work.


*By saying “no” to certain things. Maybe, back at the office, I would’ve normally said “yes,” but it was obvious that I couldn’t take any more. I was so overwhelmed that I’d say “no” to any extra things they’d ask me to do. I was already working a lot by doing the basic mandatory minimum. I was putting in an extra half hour in the evening, you know, to sit and catch up as much as I could; well, I couldn’t take any more than that. I think I also protected myself a bit, by saying “no” to certain things.*
(Eva)

*Increased use of technology to keep children entertained.* The interviewees acknowledged an increased use of technological devices to entertain their children. Despite having tried to avoid this in their children’s upbringing, women had no other choice, in this very complex scenario, but to resort to technology if they were to work from home. Very often, this decision gave rise to feelings of guilt in the women. According to Inés, for instance, these feelings subsided when she found out that other women were experiencing similar situations.


*Unfortunately, I had to throw my values away to survive work and perform well. I had to allow them many more TV hours than I would normally be happy with. To be honest, that actually stung me quite a lot, but it was the only way they would leave me alone and I could focus on my work.*
(Consuelo)


*And you can’t tell them to go play on the street, because you can’t. You have to use what you have in the house. We have TVs, tablets and mobile phones. So then, in the end, all mothers are a bit guilty. I’m letting them abuse this technology, which I don’t like. Right? But then I also saw on social media that many moms said the same. They felt the same way, didn’t they? I understood I wasn’t the only mom doing that, even though I was having a tough time.*
(Inés)

### 3.3. Is Co-Responsibility a Matter of Luck?—Challenges When Sharing Housework during Lockdown

#### 3.3.1. Childcare Remains Primarily a Female Task: School Support and Greater Mental Burden on Women

*School support.* During the lockdown, mothers had to take on the responsibility of supporting their children’s education and following their progress. The tasks proposed by schools required the parents’ continuous effort and support; parents were thus overwhelmed and could not keep up with all the assignments their children were given.


*The small one, I know he did it in good faith… But handicraft? Really? I don’t have time to sit around with my child to paint and cut and paste pictures! I can’t, I have to work. I don’t get paid to sit and paste pictures.*
(Inés)


*I remember, with my child, it was ridiculous. I wrote to his teacher, I said: “Look. We’re just not going to do certain things.” Because they’d ask you to print out the task, then you had to do it, scan it, and then send it to them over Google Classroom. My seven-year-old obviously doesn’t know how to do that, so it was me who had to do it.*
(Blanca)

*Greater mental burden on women.* Generally, there was a greater mental burden on women during the lockdown, even if they perceived their partners to be co responsible. As reported by Blanca and Emilia, it is still seen as natural for women to take on childcare roles, and this makes them approach childcare differently compared to their partners.


*I was trying to get up as early as possible. I’ve always liked getting up early, but it’s like this for women—we’ll never know if that’s because of our nature, or because we have such a long day ahead, and we have such little time for us as mothers, that maybe we just assume that—the fact that we like getting up early. But maybe we don’t.*
(Blanca)

The women we interviewed felt a greater mental burden because of the planning, managing, and organizational duties they had to assume in the household during the lockdown. In this area, they recognized a lack of co-responsibility in men, including a tendency to follow and execute instructions. Women attributed this burden to womanhood and the difficulty of detaching themselves and delegating to their partners.


*About that, I do think that he and I have a good balance, and we’re both aware. But there is still this mental burden, which is this invisible burden which lingers, and I’m sure it affects me more than it does him, because I do think this is something we have in our DNA as women.*
(Mercedes)


*I mean, everything you tell him to do, he’ll do it, but he doesn’t think for himself—zero. He doesn’t even remember to put the kids’ snacks in the bag before going to the park. Nothing, nothing at all. It’s me who goes “we have to do this and that,” “we have to wash them,” “we have to put their clothes on,” “we have to eat,” “we have to go shopping”—anything that needs thinking, it’s me who does it. Then he does what he’s told to do. But no thinking.*
(Alba)

#### 3.3.2. Co-Responsibility and Partner Conflicts

According to the women we interviewed, their partners found it more difficult to draw boundaries between productive work and their families. One possible explanation that women gave is the differential importance attributed to productive work during lockdown, as opposed to childcare. For instance, Cristina and Blanca reported misunderstandings with their partners, which made them feel frustrated and powerless. This was mostly because the men presumed that their own productive work had to be prioritized.


*At the beginning we more or less got along, but then it became maddening because work was very important, both for me and for my partner. None of us want to lose our jobs, and that’s how we got into many conflicts. Because, well, you feel that what you’re doing is very important, but so do I, my work is also very important. And that’s how we got into conflicts because I wouldn’t understand how he’d put his work before caring for his children in a situation like that.*
(Cristina)


*I remember how he said: “Oh no, no, I have to work my eight hours, I have to stay up in the studio and work my eight hours”. And I was like “Wait a minute, let’s see. I also have to work my eight hours” […]. I was very surprised when he said: “Oh no. I’m going up there and isolate myself from them.” Well, you can’t. I’ve said many times, that sentence meant a lot more. It was the lack of understanding for what I was going through.*
(Blanca)


*That was how, in such a situation, conflicts arose between partners. These conflicts were linked to establishing priorities and boundaries between productive work and sharing the housework.*



*Then the weekends came and I was like “Oh no, I don’t even want to sit with you and watch a movie, because we’ve spent all week arguing about this or that.” And then things would mellow down and we started to communicate again.*
(Consuelo)


*I think this does one of two things. It either unites you as a couple or destroys you. It either unites you, and you say “come on, let’s team up, we can do this together,” or you start fighting each other and you say “look, as soon as this is over, I’m getting a divorce.”*
(Ana)

### 3.4. Breakdown of the Care and Social Support System

#### 3.4.1. Women left Unprotected, with No External Support

Women acknowledged that they had always needed external support to be able to work and care for their children. Before the lockdown, they were supported by grandmothers and schools or nursery schools. They had relied on external help for basic household cleaning. When the strict stay-at-home order was imposed in Spain, this care support system collapsed. The main reasons were the closure of schools and movement restrictions, the consideration of elderly people as a risk group, and the limitations to contact between so-called “bubbles”.


*Not having any occasional external help. Before, my child would spend a couple of hours with his grandparents, or they would pick him up, feed him, and get him to take his nap. That kind of thing can free you up for a whole afternoon, or one morning, and of course, we no longer had that.*
(Clara)


*It’s not only having the children at home. Before, they were looked after by someone else for eight hours a day, but now you have to look after them for those hours. There were also a number of hours a week when someone could come and help you at home, but now they can’t come either. These are all things that you have to do on top of your work.*
(Eva)

The women consistently felt a lack of protection in this scenario, as they had no support measures to help them balance work and family. Childcare depended solely on the members of the core family. Some of the solutions they considered included moving in with their parents, requesting unpaid leave, using paid holidays, or reducing their contractual working hours, so they could limit work demands and devote themselves exclusively to their children. In most cases, these solutions involved lost wages and therefore, fewer resources and a potential loss of financial autonomy for the women.


*Well, clearly, when it came to balancing work and family, we were left on our own. Because there was no alternative, you had to do as you could, at home, with the kids. It was your problem. So it was very complicated. Not feeling supported [weeping]. They were saying they would give benefits to parents so they could take some unpaid time off—or something. I don’t know, they would announce these things, that maybe could have helped, but nothing came out of it in the end.*
(Marta)


*The bad thing for me was having no school. So then we made the decision. It took us two weeks, before catching up and then realizing what was happening, and then my baby—it was so demanding.*
(Carla)

It was up to individual families to tackle this breakdown of the care system, as it was not approached as a collective problem. The interviewees reflected about potential measures that could have been implemented to regulate telework better, e.g., regulating work hours and workloads to avoid abuses by companies, facilitating unpaid leave or contractual reductions, increasing workforce teams and hiring unemployed people. Likewise, childcare support measures could have been adopted to facilitate the division of care between the parents; for instance, childcare could have been delegated to the younger population or could have been labeled an essential job.


*About work, well, they could have hired other people, unemployed people, to do the work during that period, and then I could’ve gone back when the whole thing was over.*
(Irene)


*It would have been easier, during the hard part of the lockdown, if they’d let people who do housework to continue to work at other people’s houses or caring for other people’s children; if they’d let those people carry on working. For example, during the lockdown, it would have helped to have a young neighbor in the same building, boy or girl, say a 20-year-old, who could’ve come over to my place for a couple of hours and be with the children. As if it was essential care.*
(Eva)

#### 3.4.2. Invisible Care Crisis and Need for Acknowledgment

According to Blanca, the conflict that parents experienced when trying to balance work and family was insufficiently visible in the public arena, even though this conflict was obvious. For this reason, the government did not consider it a priority to approve policies and relieve the parents’ difficulties in complying with work duties and childcare at the same time.


*Maybe, this issue should have been talked about more. Nobody talked about how hard it was to work from home! When they talked about the pandemic and the people working from home, do you think they thought about people with small children?!*
(Blanca)

A specific example of this is Carla, who emphasized the need she felt for acknowledgment by her partner throughout the lockdown. Every day she would tell him about all the childcare and household tasks she had done, as a way to make him aware of the fact that someone (she) was effectively performing those tasks.


*I have a great need for acknowledgment by my partner. I need him to realize all the work I do. But I may not do it well, because I ask for it so much, I get annoying. I seem to be complaining all the time; I have this urge to say: “I did the laundry, I hung out the clothes to dry, I also made dinner, and now on top of that the children behaved really badly, and all of that during my work”*
(Carla)

### 3.5. Decline of Health by Women Trying to Balance Work and Family Life

#### 3.5.1. Mental Health Effects: A Roller-Coaster of Emotions and Stress

The strict stay-at-home mandate and the difficulties related to balancing work and family affected the women’s mental health. Our interviewees reported loss of emotional control and feelings of irritability, emotional lability, and impatience, especially with their small children. Simultaneously, feelings of guilt emerged because of all the behaviors described above.


*I noticed it especially in the way I treated my husband and my children. I knew I wasn’t talking to them like I normally do. I mean, I became numb to how to treat my children. I was totally numb as to how to treat them. I’d try to have them sit down, to do what they were told at once. I wanted them not to make things more complicated. And that was impossible. I spoke terribly to them.*
(Cristina)


*I was much more irritable. I’m usually very calm and sweet. But during that time I would snap, I would get loud, I was getting so angry. I couldn’t bear it; I had no patience with my children. I would scream at them at lot. It wasn’t pleasant […]. It was very stressful. When months went by, I started noticing it in my head, how much more sensitive I felt. I wanted to cry constantly, I felt I couldn’t cope with everything.*
(Sara)

The lockdown became a hostile environment for women in terms of mental health. Beatriz believes that she would have benefited from some form of psychological support, whereas Marisol admitted to having taken anxiolytics to feel better, endure the experience, and avoid panic attacks.


*I would have liked something, really. I know it sounds hard, but I mean something psychological. Some psychological help… I know it really sounds hard. Someone to say to me “Come on, Beatriz, this is how we’ll do it, everything is going to be alright, we’ll sort it out. Your children will be alright, everything will be alright. They won’t have trauma.” I don’t know, something, a little bit more help.*
(Beatriz)


*At some point I noticed I was getting anxiety. Well, there was a day when I had to take a Trankimazin [alprazolam] because I was like, if I don’t, I’ll have a panic attack. But I had to take it, because I could feel it, I had this sensation… Now this, then that, and that, and that…*
(Marisol)

A clear example of this situation was the emotional lability shown by Alba during her interview. She admitted that she was experiencing a complicated time because of all the emotional difficulties arising out of the pandemic.


*As soon as I begin to talk, I start crying. I guess it’s affected me. All this time. Well, generally, my lifestyle and everything, now I do notice it’s become a bit of a knot. Because this is how upset I get every time I think about it. I understand that I’m not doing very well […]. It’s like I have something inside me that snaps at certain times.*
(Alba)

#### 3.5.2. Fatigue and Stress as Part of a Routine

Some interviewees reported having felt tired because of the prolonged work hours and childcare. One of the strategies followed by women was to get up very early so they could work during the hours before their children would wake up. This shortened their sleep time and made them feel exhausted.


*Getting up at 5 a.m. to work… In the end I did it so many times! In fact, on some days I got up even earlier, at 2 a.m., because I just couldn’t sleep, and I worked till 8 or 9 in the morning… And then I would carry on all day. Obviously, I wasn’t getting any rest… I was exhausted!*
(Nerea)


*I like taking on many things, doing many things at the same time, because I’ve always done it. But right now I’m so tired […] because I’m not getting any sleep. I sleep very little. That’s the worst thing for me. I get very little sleep, but I don’t feel tired during the day. But I do feel I’m in a terrible mood.*
(Alba)

#### 3.5.3. Self-Care Deficit

Self-care for mothers was a serious challenge during the pandemic. As reported by Irene, the need to have spaces for personal care had never been a priority for her. During the lockdown, however, this was taken to the extreme, and self-care was seen as a waste of time, as in Luisa’s case. The main limitation for self-care was the lack of time, which exacerbated during the lockdown.


*Nothing. Zero. I had no time for anything. Some people were watching series, doing yoga, but I couldn’t do anything. I couldn’t read, or watch series, or do yoga. I couldn’t do anything—nothing at all. All I could do was work and combine that with the housework—taking care of my child—and nothing else.*
(Marisol)


*It was either work or the girls. There was no option to say “wait, I need five minutes alone” […] especially thinking I cannot waste any time on myself, because I’ll get behind on work. I mean, I won’t sit and enjoying reading a book if I’m lagging behind with my work. Mentally, I didn’t even allow myself that option.*
(Luisa)

Having no personal time during the lockdown, as expressed by Blanca, generated persistent distress.


*I went through a really bad week, because I had to work. That’s all I could do. I was getting so upset then, I felt I couldn’t cope. I didn’t even have one hour for myself!*
(Blanca)

In addition, it became clear that women usually put everybody else’s needs before their own.


*You eat worse, because you always leave yourself to the end. That’s true. As long as their lunch was fine, and their tea was fine, I could get by with a nibble.*
(Carla)


*It was the first thing I gave up. Having to log on to that one hour of Pilates with an on-line instructor was causing me even more anxiety. I was so anxious to log on, that eventually it was more stressful than not doing it.*
(Sara)

#### 3.5.4. Lack of Exercise and Changes in Bodily Perception

During the lockdown, the women’s lifestyle underwent several changes. They stopped doing physical activity and changed their dietary habits. This was due to the measures imposed to prevent the spread of the disease, but also because of the reality women were experiencing at home, which made them feel stressed and eat worse.


*I lost weight. I get very nervous and you can see it in my body. I got very skinny. Yes, I lost weight, because you don’t do any exercise, and I don’t enjoy food so much. Also I get so nervous, so in the end you can see it in my body, you can tell I’m not doing any exercise.*
(Luisa)


*I did, I got very fat, and normally I’m quite slim. I got very fat because of the anxiety. I guess we all get this anxiety when something so strange happens. What we went through was so hard, and all I did was just sit there, I never sit for so many hours.*
(Beatriz)

## 4. Discussion

As this study shows, during the COVID-19 lockdown, women faced the challenge of complying with both their productive awork and childcare within the same physical and temporal space. However, they did not meet their expectations—neither as mothers nor as professionals. Women faced several difficulties, including the collapse of their support system, closure of schools, telework with an overload of productive work, lack of time, and unequal distribution of duties with their partners. For these reasons, they pursued different strategies, such as working longer hours, increasing the use of technology to keep their children entertained, and setting strict boundaries in their work. Women felt unprotected and invisible, as they had no governmental support for childcare and no regulation of their telework situation. If we analyze these experiences according to the biopsychosocial model of health described by Sara Velasco [30], the consequences affected the women’s physical health (self-care deficit and fatigue), mental health (anxiety, stress, feelings of loss of control, guilt, frustration, and irritability), and social health (conflicts with their partners).

For the women we interviewed, working from home did not necessarily mean better work-family balance. Given the flexibility it accords, telework can make it easier for women to strike such a balance, but if not implemented properly, as was the case during the lockdown, it results in fragmented work hours, constant interruptions, and no real disconnection from the work environment [31,32,33,34]. A study conducted in Costa Rica [35] emphasized that telework does not necessary involve flexibility; certainly, if employers engage in controlling behaviors, employees can feel just as constrained as when working in an office. According to the results of our study, bosses commonly engaged in such behaviors, demanding ongoing availability of women as proof that they were indeed working.

The lack of technological resources and ergonomic workstations to maintain privacy from other household members was also reported in another study in the United States [36], which suggests that the difficulties experienced by women were similar across countries. Men were more successful in securing a separate professional space within the home; this could be explained by normalized gendered assumptions within the couple such as: the man’s productive work is more important; the woman needs to stay within the domestic space to keep the household under control [37]. In other words, gendered power relations between men and women also determine the use of spaces within the home, which was crucial when teleworking to establish limits between one sphere and the other.

Failure to meet the expectations of women as both mothers and professionals is also described in a study conducted among female academics, for whom the main challenge was finding a balance between their children’s needs and their careers [38]. Employers still understand the ideal worker as one who has no family responsibilities [16]. To be able to accomplish work demands, women had to increase the use of screen time to keep their children entertained. The feelings of guilt linked to this strategy are consistent with those noted in another study conducted in the United States [39], in which women underscored it as the sole viable strategy to manage full-time work and full-time parenting. Another strategy used by women was extending their work hours—both to adapt to their children’s timetables and to compensate for missed work hours during the day [38]. This is confirmed by the results of the Eurofound 2020 survey [40]: 18.2% of Spanish respondents reported an increase in their working hours during the pandemic; 37.3% worked in their free time to be able to meet work demands; and 58.4% could not devote as much time as desired to their families. On the contrary, international studies [41,42] have concluded that it was mostly women who reduced their working hours or even stopped working temporarily during the lockdown—but we did not find this in our study.

During the lockdown, the women interviewed in our study took on a greater load of housework and childcare than did their partners. The literature [43,44,45] suggests that gender inequalities in the division of work were maintained during the lockdown, and their consequences will be long-lasting. The results indicate that traditional gender roles re-emerged during the pandemic, with women shifting closer to the domestic sphere and away from their professional autonomy [8,12,37]. This inequitable division of labor can be explained by the naturalization of women as carers [44,46]. This may explain the findings of our study regarding the importance and priorities given by men to their own productive work.

Women felt stressed and overwhelmed when they had to take on the duties of head of household or household manager. The mental load is a form of cognitive and emotional labor taken on by women without negotiation, as a kind of boundaryless unpaid job that intrudes into other areas of their lives and affects their health and wellbeing [47]. Supporting their children’s education was a big challenge for mothers, who had to play the role of teachers and combine their children’s remote learning with their own duties [48]. Consequently, the mental health of mothers was affected by this new responsibility. It increased their levels of stress, anguish, tension, and loss of control [48,49,50]. There is evidence that the unequal distribution of labor and the exacerbation of traditional gendered roles during the lockdown impacted women’s health, especially their mental health: increased levels of stress and decreased personal satisfaction led to frustration and exhaustion [38,51]. In our study, we have also observed uncontrolled emotions and impatience, irritability, distress, mental and physical exhaustion, and emotional lability, which was clearly reflected in some of the interviews. A perceived lack of support for childcare is related to higher levels of depressive symptoms in women [11]; conversely, satisfaction with how labor is shared with their partners appears to be protective against stress and mental health effects, enhancing women’s health and wellbeing [17,44].

Women put the needs of other family members first, before their own. Studies conducted in other countries have also shown that women tended to neglect healthy lifestyle habits related to diet and physical exercise, which in some cases, caused changes in their perceived bodily image [18,43]. The establishment of priorities at the expense of their own wellbeing is justified by the feelings of guilt that women harbored for becoming “bad moms”, which eventually left them mentally and emotionally exhausted [45].

*Limitations and strengths.* The following limitations should be taken into account: (1) the study population is restricted to heterosexual women living with their male partners, and most of the women had attained higher education and held formal jobs, so whether the results can be transferred to other similar contexts should be evaluated; (2) we did not take into account men’s statements, so the study only shows the women’s subjective perception of their partners’ behavior; and (3) during the recruitment phase of the study, women had little available time; this was the main reason why some women refused to take part, even though they were interested in the study. Although the results are not statistically generalizable, they are useful for theorizing about the experience of working mothers during confinement. An effort was made to contextualize the results, so that readers could better assess the applicability to other fields of study.

Regarding the strengths of our study, we followed the reliability criteria described by Lincoln and Guba [52]. Following an emergent design, the data were analyzed in parallel to data collection. In addition, field notes were taken during the interviews, especially of the interviewees’ non-verbal cues, as they exhibited emotional lability on several occasions over the course of the interview, providing additional valuable information. All five investigators contributed to the study design and the final interpretation of the results.

Future lines of research could delve into the challenges faced by single mothers and women in essential jobs who could not remain at home and were required to be physically present at work.

## 5. Conclusions

During the lockdown, women were forced to choose between meeting their roles as mothers or as professionals. This exposed them to both internal and external demands which eventually, had damaging effects for their health. The division of labor with their partners was unequal during the lockdown. Women tended to spring back into traditional gendered roles—which some families had already left behind. The consequences for the health of mothers who contended with work-family conflicts during the early strict phase of the lockdown in Spain included stress, fatigue, anxiety, distress, loss of emotional control, irritability, emotional lability, partner conflicts, and a deficit of self-care, among others.

Our study unveils a problem which was kept hidden during the lockdown, namely how women tackled a care crisis during the COVID-19 pandemic. Strict home confinement measures in Spain made the work-life balance problem even more difficult for women. In a future health crisis, it would be recommendable to consider the outdoor leisure needs of families with young children for their well-being and mental health.

Lack of work-family balance can negatively impact women’s health and can reverse progress in gender equality. On one hand, awareness of this problem should be raised among governments and employers, and public policies should be improved and implemented to facilitate work-family reconciliation and co-responsibility within couples. On the other hand, health professionals should be aware of the needs and challenges experienced by women with children in situations of crisis or health emergencies. Specifically, this should enable them to develop family and community health strategies and interventions in order to promote health and wellbeing.

## Figures and Tables

**Table 1 ijerph-20-04781-t001:** Participants’ social and demographic characteristics.

Pseudonym	Age	No. of Children	Job Sector	Currently Teleworking	Full- or Part-Time Work
Nerea	41	2	Public	No	Part time
Carla	41	2	Private	No	Part time
Clara	41	1	Public	Yes	Full time
Ana	42	1	Private	No	Full time
Luisa	40	2	Public	No	Full time
Beatriz	44	2	Public	No	Full time
Loreto	40	3	Private	No	Part time
Eva	38	2	Public	No	Part time
Irene	42	2	Private	Yes	Full time
Cristina	37	2	Private	Yes	Full time
Consuelo	35	1	Private	No	Full time
Marisol	54	1	Public	No	Full time
Mercedes	38	4	Private	No	Full time
Emilia	42	1	Private	No	Full time
Marta	36	2	Private	No	Part time
Blanca	42	2	Public	Yes	Full time
Sara	41	2	Private	Yes	Full time
Inés	42	2	Private	Yes	Full time

**Table 2 ijerph-20-04781-t002:** Interview script for the semi-structured interviews.

Questions
(1) How did your family routine change during the first COVID-19 lockdown (14 March to 21 June 2020)? How is that family routine developing under the new normal (21 June onwards)?
(2) How did you organize yourself working from home? (Effort to adapt, stress, complying with work hours, workdays…).
(3) How did you and your partner organize yourselves working from home? (Overlapping working hours, strategies adopted…).
(4) What are the main challenges and barriers to working from your home?
(5) How did you perceive the level of demand at work during that period of the lockdown? What about now? What impact did that level of demand have on your wellbeing? What strategies did you use to comply with those demands at work?
(6) How do you think your work-family balance was affected by the lockdown and now by the new normal? What were, for you, the main challenges to balancing work and family during the pandemic?
(7) How did you feel, in terms of health, during the lockdown, and how do you feel now under this new normal? Can you please provide details about physical, psychological and social aspects? (Relationship with your friends, your partner, productive work…).
(8) What kind of self-care did you engage in during the lockdown? What about under the new normal?

**Table 3 ijerph-20-04781-t003:** Themes, categories, and quotes.

Theme	Category	Quotes
Telework—Characteristics and challenges of a new labor scenario		“We started to work remotely with no planning. We had no choice.”
Survival and chaos—Inability to work, look after children and manage a household at the same time	Constantly combining work and childcare within the same physical and temporal space	“It’s not realistic to do a full double workday.”
	Strategies adopted to survive the lockdown routine	“Some children were just glued to the TV all day so their parents could work from home.”
Is co-responsibility a matter of luck?—Challenges when sharing housework during lockdown	Childcare remains primarily a female task: school support and greater mental burden on women	“It’s always us—women—who end up losing. We take on more workload.”
	Co-responsibility and partner conflicts	“I have to lock myself in and work my eight hours.”
Breakdown of the care and social support system	Women left unprotected, with no external support	“We were left on our own. Women had to take on the childcare on top of their professional work.”
	Invisible care crisis and need for acknowledgment	“If you want to bring up your children and work at the same time, you need help.”
Decline in health by women trying to balance work and family life	Mental health affected: a roller-coaster of emotions and stress	“Emotionally, I could have exploded at some point.”
	Fatigue and stress as part of the routine	“There’s no time to switch off, and a lot of stress to get everything done.”
	Self-care deficit	“If you say that you want to take care of yourself, they’ll look at you as if you were a freak.”
	Lack of exercise and changes in bodily perception	“You eat worse, because you’re always last on the list.”

## Data Availability

The data belong to a wider unpublished Ph.D. study. It will be available upon request from the authors.

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
