# Peer review of "Juggling during Lockdown: Balancing Telework and Family Life in Pandemic Times and Its Perceived Consequences for the Health and Wellbeing of Working Women"

_ijerph, 2023, doi:10.3390/ijerph20064781_

Round 1

Reviewer 1 Report

This is a very interesting and relevant study. I enjoyed the writing very much and felt it truly captured the complexity of the telework and how it had affected female professionals differently. I do have a number of minor suggestions: 

1. The analysis for section 3.1 needs more clarity and logical consistency. If the research question is about pandemic-specific challenge, then the content was not focused well and seemed to capture more of the transitioning period (reopening, returning to work). 

2. Line 222 (page 6) "complicity and difficulties for teamwork" were not really "strategies," right?

3. Among the people you have interviewed, there are meaningful number of people who had worked and/or continued to work remotely. But in your analysis, this factor was not addressed. Specifically, is the phenomenon you have captured a result of a temporary pandemic era arrangement or a typical telework effect? I would think this has important practical implications whether it is a temporary situation or more permanant one. 

4. It would be helpful for the reviewers to have a list of the scripted questions you have used for the interviews. 

Reviewer 3 Report

Thank you for giving me the opportunity to read your paper. Here are my two main comments:

1. Please expand your introduction to lead the reader to justify your methodology, data collection, need for the study, and the purpose of your study.

2. Authors need to clarify/justify the use of the mentioned methodology,  software, and data analysis.

3. looking at the past references, I suggest authors follow the previous studies in reporting the findings with clear tables and content.

4. in its present form, the manuscript seems over simplified.

Round 2

Reviewer 3 Report

The authors have revised the paper satisfactorily.